# Threshold analysis regarding the optimal tax rate and tax evasion. Empirical evidence from Taiwan

Yu kun Wang 👤*, Li Zhang

Department of Economics and Finance, Zhanjiang University of Science and Technology, Guangdong, China

* WYK2@ulive.pccu.edu.tw

## Abstract

For a long time, governments of all countries have attached great importance to the development of underground economic activities. The reason is that the characteristics of the underground economy are hidden and the information disclosure is not sufficient, which not only distorts the economic data indicators, but more importantly, the existence of the underground economy has led to the loss of a large amount of tax base, affecting the long-term economic development of the country. Whether raising the tax burden rate boosts the tax revenue or expand the scale of the underground economy. In this paper, we use Kuznet Tax Curve (KTC) method to analyze the relationship between GDP and TTR/DTR/ITR. We find that the tax base erosion rate of indirect tax is lower than that of direct tax. In addition, we explore the relationship among economic growth, tax rate and tax revenue and adopt SUR-OLS method and Threshold approach to estimate the response of economic growth on total tax revenue(TTR), direct tax revenue(DTR) and indirect tax revenue (ITR) in Taiwan from 1991-2020. Our empirical research shows that when DTR tax rates are between 12.59% and 13%, an increase in income leads to a decrease, not an increase, in DTR, leading to severe tax base erosion. That is, the relationship between GDP and DTR presents a N-shaped relationship. However, ITR does not exist any tax rate threshold effect. Obviously, with the increase of GDP, ITR also increases. This reflects that the difference of tax structure between direct tax and indirect tax plays a key role in the relationship between tax rate and tax base erosion.

## 1. Introduction

In the 1950s, Kaldor [1] and Cagan [2] mark the beginnings of preliminary research of hidden economic activity. Since then, more and more literatures have focused on discussing the relationship between undeclared income and tax erosion. Smith [3] defines the UE as the production of market-based goods and services, whether legal or illegal, escapes detection in the official GDP estimates. The formal economic theory of tax evasion can be dated to Allingham and Sandmo [4] who pointed out that when the tax burden rate increases, taxpayers must make a measurement between the expected benefits of tax evasion and risk bearing. Welch

**Data Availability Statement:** All relevant data are within the paper and its Supporting information files.

**Funding:** The author(s) received no specific funding for this work.

**Competing interests:** The authors have declared that no competing interests exist.

and Goyal [5] and Kostakis et al. [6] demonstrate that predictability may be time-varying and that the impact of predictors may be evolving over time. In general, there are two common modeling tools to deal with parameter instability in PR models: structural change and threshold model. In this study, we took Taiwan as a case study to explore the relation between TTR/DTR/ITR and GDP over the period from 1991 to 2020. Concurrently, we adopt Hansen's approach to measure the size of tax base erosion over the same period using Tax Kuznet Curve (TKC) approach and select the parameter "tax burden rate" as a threshold variable to capture the response of income variation upon tax revenue. Our paper is organized as follows. In section 2, reviews relevant literature on underground economy and tax evasion, section 3 deals with the methodology and model, section 4 differs from existing empirical methods, we take Taiwan as a study case, using Kuznet Tax Curve (KTC) approach and the SUR-OLS method to calculate the gap between the actual tax revenue in 2020 and estimated tax revenue, then we acquire the amount of tax base erosion in 2020. Further, we estimate the relationship between TTR/DTR/ITR and GDP. Section 5, referring to Wu et al. [7], we construct threshold analysis model and take Taiwan as an study case to propose an empirical analysis and discuss the tax burden rate threshold effect for taxpayers and its reaction on TTR//DTR//ITR. Section 6 recapitulates concluding remarks and outlines policy implications.

## 2. The literature review

According to Schneider and Enste's [8] survey, during the last decades the underground sector was nearly three-quarters of the officially recorded GDP in Nigeria and Thailand, but it amounted to a noteworthy 15% in the OECD countries as well. Pissarides and Weber [9] estimated the underreporting of income by using the data of the household food expenditure survey in the United Kingdom. The study found that the real income of self-employed households was 1.55 times the declared income, and then estimated that the size of the British underground economy accounted for about 5.5% of GDP. In addition, Johansson [10] studied the income leakage of Finnish households and found that the income under reporting rate of self-employed households was 25% to 30%. Milorad and Williams [11] indicate that 22.6% of all employees in Montenegro are unregistered employees. In addition, 17.5% of all formal employees received under reported salaries from their employers in order to avoid paying taxes. Wang et al. [12] use a cash deposit ratio approach and a currency demand approach to estimate UE size.

In addition, Giles et al. [13] depict that an increase in the effective tax rate has a greater effect on the UE than a decrease in New Zealand. Zhiqin and Qunli [14] outline that tax burden rate is positively related to the underground economy. In addition, Bhattacharyya [15] finds clear evidence for the U.K. (1960–1984) that the UE has a significant effect on the consumer expenditure. Another, the night light images are taken by the operational line scan system (OLS) carried by the US military meteorological satellite program (DMSP) from 1992 to 2013. When the data were released, the abnormal lights, background noise and other non urban lights had been eliminated and could be directly used for relevant research. Similarly, Elvidge et al. [16] propose in 1996 that there is a strong correlation between night light and population, GDP and power consumption data. However, NOAA has no night lighting data of Taiwan from 1992 to 2013. The above is the situation of tax base erosion in some countries.

Similarly, Schneider and Enste [8] point out, at least two thirds of the income earned in the shadow economy is immediately spent in the official sector, revealing UE and the official sector might thus be complements. Specially, Hindriks et al. [17] consider that some tax payers may collude with the inspector so the inspectors underreport the tax liability of the tax payers in exchange for a bribe. However, Jorge and Mark [18] denote that the effect of increased

enforcement effort in a given mode has an ambiguous effect on compliance in the targeted mode as well as the untargeted mode. Li Yiting and Tang Ruyin [19] pointed out that high inflation will erode the purchasing power of cash, so people will tend to hold less cash, reducing the incentives for people to engage in underground economic activities. Another, Giovanni et al. [20] propose the effect on evasion and government revenue of two policy instruments: a tax on cash withdrawals (TCW) and a tax rebate conditional on having the receipt. Their research shows the tax rebate reduces evasion but it is costly if tax evasion is low. Din [21] considered the heterogeneity of tax sources and found a positive relationship between the personal tax (direct tax) rate and the underground economy from the data of Malaysia, but a negative impact on the sales tax (indirect tax) rate.

In addition, Cranor et al. [22] analyze a large field experiment conducted with the Colorado Department of Revenue to study the presentation of financial incentives and social norms in tax delinquency notices, their research suggests that attention to seemingly minor decisions about the wording of notices sent by tax authorities can increase tax payments and reduce administrative costs associated with taxpayer delinquency. Advani [23] described that tax compliance varies with personal characteristics, and male and young people are more disobedient. Friedman et al. [24] found in a transnational analysis that higher tax rates are related to less underground economies. However, the empirical results show that this relationship is not very stable. It is worth mentioning that most of the domestic and foreign current literatures are discussed in the linear model, and the possible nonlinear relationship between them has seldom been discussed.

According to the traditional research, it is assumed that the impact of GDP on tax revenue exists nonlinear characteristics due to different degrees of tax rate. In order to verify this nonlinearity and alleviate the potential endogeneity of the traditional regression model. Granovetter [25] and Granovetter & Soong [26] propose the threshold model. In the spirit of Granovetter's threshold model, the "threshold" is the number or proportion of others who must make one decision before a given actor does so. In addition, Bick [27] applies non-dynamic (static) panel threshold regression that propounded by Hansen [28] on a balanced panel data from 40 developing countries. Kremer's [29] findings reveal a threshold inflation of 2.53% for industrial countries and 17.22% for nonindustrial countries. showing the relationship is significantly positive below the threshold and significantly negative above the threshold for the industrial countries. Gonzalez et al. [30] consider a nonlinear panel model which is called the panel smooth transition regression (PSTR) model. Their research generalized the PTR model by allowing the regression coefficients to change smoothly when moving from one "extreme" regime or state to another. However, the PTR model separates the observations into several sets or groups based on the value of the threshold variable with sharp "borders" or thresholds.

In recent research, Baumann et al. [31] design a variant of an optimal stopping task that allowed people to quantitatively characterize the deviations of human behavior from optimality and found that humans apply a simplifying strategy, where thresholds are linearly increased over time. Tariq et al. [32] examine the nonlinear relationship between financial development and economic growth in Pakistan using the threshold regression model for the period 1980–2017. They research indicates that economic growth responds positively to financial development when the level of financial development surpasses the threshold value of 0.151. However, when financial development lies below the threshold value (that is, 0.151), its impact on economic growth is negative. Yang et al. [33] extend Hansen's [34] constant threshold regression model by allowing for a time-varying threshold which is approximated by a Fourier function. Least-square estimation of regression slopes and the time-varying threshold is proposed, and test the existence of threshold effect and find there is little efficiency loss by the allowance for

Fourier approximation in the estimation procedure even when there is no time-varying feature in the threshold.

In addition, Belarbi et al. [35] adopt Buffered threshold panel data model(BTPD) to examine the combined effects of oil dependence and the quality of institutions on economic growth. To do so, they introduce a new buffered threshold panel data model and apply it to 19 oil rent-dependent countries over the period 1996–2017, their research show that the relationship between growth and oil-dependence is not linear. Another, Zhang and Kim [36] establish a model of FDI location and explore to examine the threshold role of institutional quality in determining the relationship between labor costs and FDI location, using data from 14 South and Southeast Asian countries during 2000–2017, evidence shows that effects of labor costs on FDI are nonlinearly decreasing because their institutional quality is improved above threshold values. Zhiqi's [37] research contributes to the literature by distinguishing small-scale taxpayers from general taxpayers in terms of the optimal sales threshold for VAT. their research analyzes how the optimal sales threshold varies with changes in administrative and compliance costs and in tax rates. Furthermore, Yan et al. [38] use the threshold model to analyze the nonlinear characteristics between PSA and $CO_2$ emissions under different degrees of government intervention. Similarly, Yu and Fan [39] requires that the variables affecting threshold are given or predetermined, but, in most applications, it is difficult to explore the factors which affect the threshold value in advance. Wang et al. [40] construct a threshold effect model, sets the institutional environment as the threshold variable, and empirically analyzes the impact of Internet development on the supply efficiency of government public services.

There are other literature on discussing Taiwan's underground economy. Wang et al. [12] examine the asymmetric response of the underground economy in Taiwan to the fluctuation of tax rate and measure the UE size from 1962 to 2003 using cash ratio approach and currency demand approach and find an increase in indirect or direct tax has a greater effect than the corresponding decrease. Ho and Tsai [41] examine the difference in the impact of different tax sources on the scale of Taiwan's underground economy, and found that business tax had a significant positive relationship with the underground economy, while income tax had a slight positive relationship. Lin et al. [42] use the OLS regression method based on Bai and Perron [43] to analyze and obtain the endogenous threshold of tax burden rate and discuss how the tax burden rate affects Taiwan's underground economy.

Our paper differs from the traditional literature, we adopt the threshold regression model to obtain the endogenous tax burden rate to explore the relationship between GDP and TTR/DTR/ITR. Table 1 lists recent relevant documents on Taiwan's underground economy article.

**Table 1. Literature review of Taiwan's underground economy in recent years.**

| Taiwan related literature | Analysis period | Estimation method | Genetic variable and influence direction |
|---|---|---|---|
| Dai and Sun (2003) | 1962–2002 | MIMIC model | Tax burden(+), |
| Wang et al. (2006) | 1961–2003 | MIMIC model | Tax burden(−), |
| Wang et al. (2012) | 1962–2003 | OLS model | Direct tax + (−) Indirect tax − (−) |
| Ho and Cai (2014) | 1961–2012 | OLS model | Tax burden(+), |
| Lin Zhensheng(2020) | 1976–2016 | OLS model | Tax Burden(−), |

Note: 1. The positive and negative signs in brackets represent the direction of the influence of genetic variables on the underground economy. 2. Among the tax rate variables adopted by Wang et al. (2012), direct tax+ and direct tax—respectively represent the increase and decrease of direct tax rate, and indirect tax+ and indirect tax —respectively represent the increase and decrease of indirect tax rate

## 3. Methodology, model

### 3.1. Methodology, hypothesis

The Laffer curve is a threshold effect that describes the "inverted U-shaped" relationship between tax rates (tax burden rates) and total government revenue, inspired by the above literature and theory, this paper follows the threshold model setting of Bai and Perron [43] and takes endogenous variable" tax rate" as the turning point of interval change to estimate the response of Taiwan's economic growth to total tax, direct tax and indirect tax from 1991 to 2020. That is to say, different intervals in the model are divided by threshold variables greater than a certain threshold. The hypothesis of this paper is to use the threshold method to analyze, endogenously explore whether there exists a threshold value of the tax burden rate that alters the relationship between the total tax/direct tax/indirect tax. However, testing this hypothesis requires the estimation of a non-linear model. One traditional method that solves this type of non-linearity and heterogeneity is estimating a panel threshold regression (PTR), developed by Hansen [28]. The PTR assumes that analogous individuals should belong to one group. Thus, one can divide the individuals in the sample into several groups based on observables. But this is not the focus of this article.

Due to the relationship between tax revenue and tax burden rate (tax rate). However, compared with the previous literature, most of them discussed the relationship between tax rate and underground economic size from a linear model. In this paper, we refer to the threshold model framework of Hansen [34] and Odedokun [44], selecting the tax rate as the threshold variable to explore whether there exists a threshold effect of tax rate on tax revenue, and whether the effect of dependent variables on tax revenue is different under high and low tax rates. Since the SUR-OLS method estimates the parameters of all equations simultaneously, so that the parameters of each single equation also take the information provided by the other equations into account. In general, the SUR-OLS estimates are consistently better than the OLS (equation-by-equation). Furthermore, the SUR-OLS estimator takes the correlation between the error terms into account, hence, SUR-OLS is a robust methodology for predicting (Cadavez & Henningsen [45]). As is well known, Taiwan's inland have convenient transportation links, taxpayers live in the same environment of tax laws and regulations. Hence, it has the heterogeneity of variance, and the residual has the characteristics of contemporaneous correlation. In view of this, in order to reduce the standard error, this paper uses "seemingly unrelated regression" (SUR-OLS) to test and analyze.

### 3.2. Model

The analysis of the EKC seeks to confirm whether wealth accumulation stimulates environmental degradation or contributes to improving its quality (Kaika & Zervas [46]). According to this approach, if GDP per capita is less than the level of the turning point, wealth accumulation contributes to environmental degradation; conversely, if GDP per capita is higher, environmental quality improves. In this setting, our research sets a theoretical model of the inflection point of Tax Kuznets curve(TKC) as follows. Eq (1) describes the indirect utility between tax burden and economic growth. We assume that utility function is separable in these two arguments, R and T, with the additive-separable function and additive preferences. Such that:

$$V(\boldsymbol{R}, \boldsymbol{T}) = \boldsymbol{s_1} - \boldsymbol{s_2} \times \boldsymbol{e^{\frac{-R}{\delta}}} - \boldsymbol{\gamma} \times \boldsymbol{T} \tag{1}$$

In Eq (1), $s_1, s_2, \gamma, \delta > 0$, where $s_1$ is coefficient, $s_2$ reflects the impact of real income on utility, γ reflects the impact of tax burden on utility, F represents the government's subsidy to

taxpayers below a certain income threshold or tax exemption threshold, $\tau_m$ represents marginal tax rate system, $\beta$ is income declaration rate of taxpayers, R denotes the level of income, T is the tax burden. Hence, we set the tax burden paid by the taxpayer can be expressed as

$$T = -F + \tau_{mt} \times R(\beta) \tag{2}$$

Hence, the higher the income declaration rate of taxpayers, $\beta$, the greater the T. Consider the character of progressive income tax rate system, we adopt the sustained- growth version of Guo and Lansing's's [47] nonlinear tax structure and postulate $\tau_t$ as

$$\tau_t = 1 - \eta \left( \frac{R_t^*}{R_t} \right)^\theta \tag{3}$$

In Eq (3), $R_t^*$ denotes a benchmark level of income that is taken as given by the representative household. In our model with endogenous growth, $R_t^*$ is set equal to the level of per capita output on the economy's balanced growth path (BGP), where $\frac{R_t^*}{R_t} = \theta > 0$, for all t. Hence, the marginal tax rate $\tau_{mt}$, defined as the change in taxes paid by the household divided by the change in its taxable income which is given by

$$\tau_{mt} = \frac{\partial(\tau_t R_t)}{\partial R_t} = \tau_t + \eta\theta \left( \frac{R_t^*}{R_t} \right)^\theta \tag{4}$$

where $0 < \tau_t, \tau_{mt} < 1, \frac{R_t + F}{R_t(\theta)} \geqq \tau_{mt}$, as mentioned, R represents real income, T denotes tax burden for people, reflecting the adverse impact of tax burden on the people's indirect utility. Moreover, we assume that the marginal disutility of tax burden remains unchanged. In order to eliminate the impact of structural effects, we suppose that only one commodity model is used for analysis. In this situation, firms produce aggregate output, Y, we set a constant returns to scale technology of the Cobb-Douglas type. Therefore, a country's incomes Y is expressed as Eq (5):

$$Y = P \times \lambda \times T^\alpha \times F(K, AL)^{1-\alpha} \tag{5}$$

In Eq (5), $\lambda$ is the conversion coefficient, P represents the commodity price, with $\lambda \in (0,1)$. F (K,AL) denotes aggregate production function, where K denotes aggregate physical capital and L represents aggregate labor employed in production, A represents the technical level, with A > 0, $\alpha \in (0,1)$. Eq (6) reflects the value of marginal tax burden upon taxpayers equal to the demand of reverse tax burden, which is given by:$\Gamma$

$$\Gamma^D = \alpha \times P \times \lambda \times T^{\alpha-1} \times F(K, AL)^{1-\alpha} = \frac{\alpha}{T} \times Y \tag{6}$$

Also, the value of marginal tax revenue levied by government can be expressed as follows.

$$\Gamma^s = -\frac{V_T}{V_Y} = \frac{\gamma \times \Omega(P) \times \delta}{s_2} \times e^{\frac{R}{\delta}} \tag{7}$$

Through the supply-demand production function, the expression of the Kuznets curve can be obtained through Eqs (6) and (7)

$$T^* = \frac{\alpha \times S_2 \times R}{\gamma \times \delta} \times e^{\frac{-R}{\delta}} \tag{8}$$

Furthermore, the following formula can be obtained by calculating the derivative of optimal tax revenue/burden T.

$$\frac{dT}{dR} = \frac{\alpha \times S_2}{\gamma \times \delta} \times \left( e^{\frac{-R}{\delta}} - \frac{1}{\delta} \times R \times e^{\frac{-R}{\delta}} \right) = \frac{\delta - R}{R \times \delta} \times T \tag{9}$$

Clearly, the inflection point of tax burden is $R = \delta$. This shows that when economic growth reaches a certain level, there will be tax base erosion. This means that people begin to evade taxes in an attempt to reduce their tax burden. Eq (9) is a convergence function, its value is greater than zero. That is, if n positive convergence functions are added together, the function obtained should also be convergent. Based on the theoretical models derived from Eqs (1) to (9). we seek to use empirical analyses to discuss the existence of TTR/DTR/ITR-to-GDP ratio/ Kuznets Curve and further discuss whether the Kuznets Tax Curve/ TTR/DTR/ITR-to-GDP ratio exists in Taiwan covering the 1991–2020. Obviously, if these Kuznets curve does not exist, revealing that with economic growth, tax revenue will also increase.

## 4. Data, empirical analyses

### 4.1. Empirical analyses between TTR/DTR/ITR and GDP

In this paper, we take Taiwan as a case study and use Simultaneous equations model and SUR-OLS approach to exploit the cointegration relationship among the GDP, variables TTR, DTR, ITR for Taiwan over a time period ranging from 1991 to 2020. To capture the synchronous correlation between heterogeneity and residuals in the model, our research employs SUR-OLS approach to measure the correlation among those variables, determining whether the stochastic component contains a unit root or not. The results of unit root tests are presented in Table 2, which demonstrates that all the variables appeared stationary at the first—differenced form under 5% significant level, depicting the logged variables are I(1). We next utilize the SUR-OLS regression method evaluating the residual term and estimate whether the residual term conforms to no sequence autocorrelation.

We then adopt Johansen Cointegration to test whether there exist a long-term equilibrium relationship between TTR/DTR/ITR-to-GDP. In Table 3, Trace test result shows that there exists a set of cointegrating vectors at the 5% level, and Max-eigenvalue test also indicates the same result.

Owing to the Q-statistic proposed by Box and Pierce [48] is rather weak in large samples, Ljung-Box [49] proposes another modified Q-statistic suitable for small samples. However, Box & Jenkins [50] consider that it is necessary to diagnose whether the parameters have overfitting and also confirm whether the residuals have serial correlation. Below, the results of Ljung-Box Q test are shown in Fig 1, which reveals the probability values of Q-statistics from

**Table 2. Performance of unit root test.**

| variable | N-st difference | (C,T,K) | DW | ADF | 5% | 1% | Result |
|---|---|---|---|---|---|---|---|
| TTR | 1 | (C,n,7) | 2.09 | -7.31 | -3.58 | -4.32 | I(1)*** |
| DTR | 1 | (C,n,7) | 2.05 | -5.91 | -3.58 | -4.32 | I(1)*** |
| ITR | 1 | (C,n,7) | 2.19 | -9.35 | -2.97 | -3.68 | I(1)*** |
| GDP | 1 | (C,n,7) | 1.97 | -5.17 | -2.97 | -3.68 | I(1)*** |

Note: (C, T, K) indicates whether the test formula contains constant term, time trend and number of lag periods using AIC. Standard errors in parentheses:

*** denotes the 1st- differenced form passes the stability test at 1% significance level,

** denotes the 1st- differenced form passes the stability test at 5% significance level.

**Table 3. Performance of Johansen Cointegration test, TTR/DTR/ITR-to-GDP.**

| 1991 to 2020 | | | |
|---|---|---|---|
| **H0 H1** | **Statistic** | **5% critical value** | **Prob\*\*** |
| I.TTR-GDP | | | |
| Trace test | | | |
| None* | 18.0636 | 20.2618 | 0.0976 |
| At most 1* | 3.61459 | 9.16454 | 0.4725 |
| γ = 0 γ≧1 | | | |
| Max-eigenvalue test | | | |
| None* | 14.44902 | 15.8921 | 0.0831 |
| At most 1* | 3.61459 | 9.16454 | 0.4725 |
| γ = 0 γ≧1 | | | |
| II.DTR-GDP | | | |
| Trace test | | | |
| None* | 24.5416 | 20.2618 | 0.0121 |
| At most 1* | 7.6901 | 9.1645 | 0.0945 |
| γ = 0 γ≧1 | | | |
| Max-eigenvalue test | | | |
| None* | 16.8515 | 15.8921 | 0.0353 |
| At most 1* | 7.6901 | 9.1645 | 0.0945 |
| γ = 0 γ≧1 | | | |
| III.ITR-GDP | | | |
| Trace test | | | |
| None* | 24.5416 | 20.2618 | 0.0121 |
| At most 1* | 7.6901 | 9.1645 | 0.0945 |
| γ = 0 γ≧1 | | | |
| Max-eigenvalue test | | | |
| None* | 16.8515 | 15.8921 | 0.0353 |
| At most 1* | 7.6901 | 9.1645 | 0.0945 |
| γ = 0 γ≧1 | | | |

Notes: γ denotes number of cointegrating equations; Trace test indicates 1 cointegrating eqn(s) at the 0.05 significance level. Max-eigenvalue test indicates 1 cointegrating eqn(s) at the 0.05 significance level.

the first period to the sixteenth period are all significantly greater than the 5% significance level. On the other words, the residuals estimates of model 1 to model 3 in Table 4 have no sequence autocorrelation.

The Ljung Box verification results can be seen from Fig 1(a)–1(c), the Ljung Box test statistics from Phase 1 to Phase 12 were all 5% higher than the significant level, indicating that there is no autocorrelation among the three variables, TTR, DTR, ITR in the residual items from Phase 1 to Phase 12.

We next exploit the Histogram-Normality test and Heteroscedasticity test. In Table 4, we use Breusch-Pagan-Godfrey to diagnose residual heterogeneity, which show the p-values of F-statistic, OBS * R-squared and Scaled explained SS of all models are all significantly greater than 5%, denoting that the residuals from model 1 to model 3, in Table 4, do not exist residual heterogeneity.

Note that in Table 4, the p-values of F-statistic, OBS * R-squared and Scaled explained SS of model 1 to model 3 are significantly greater than 5%. In Table 4, model 1 to model 3 correspond to the three models in Table 5 in an orderly way. Standard errors in parentheses: ***

Sample (adjusted): 1993 2020
Q-statistic probabilities adjusted for 2 dynamic regressors

| Autocorrelation | Partial Correlation | | AC | PAC | Q-Stat | Prob |
|---|---|---|---|---|---|---|
| | | 1 | -0.040 | -0.040 | 0.0504 | 0.822 |
| | | 2 | -0.129 | -0.131 | 0.5860 | 0.746 |
| | | 3 | -0.115 | -0.129 | 1.0304 | 0.794 |
| | | 4 | -0.127 | -0.162 | 1.5933 | 0.810 |
| | | 5 | 0.058 | 0.005 | 1.7169 | 0.887 |
| | | 6 | -0.053 | -0.112 | 1.8252 | 0.935 |
| | | 7 | 0.167 | 0.138 | 2.9459 | 0.890 |
| | | 8 | 0.170 | 0.170 | 4.1646 | 0.842 |
| | | 9 | -0.208 | -0.161 | 6.0848 | 0.731 |
| | | 10 | -0.068 | -0.030 | 6.2987 | 0.790 |
| | | 11 | 0.085 | 0.134 | 6.6559 | 0.826 |
| | | 12 | -0.012 | -0.041 | 6.6636 | 0.879 |

* the results of Ljung-Box Q test of variable TTR

(a) TTR

Sample (adjusted): 1993 2020
Q-statistic probabilities adjusted for 2 dynamic regressors

| Autocorrelation | Partial Correlation | | AC | PAC | Q-Stat | Prob |
|---|---|---|---|---|---|---|
| | | 1 | -0.044 | -0.044 | 0.0605 | 0.806 |
| | | 2 | -0.253 | -0.255 | 2.1209 | 0.346 |
| | | 3 | 0.046 | 0.022 | 2.1925 | 0.533 |
| | | 4 | -0.133 | -0.208 | 2.8140 | 0.589 |
| | | 5 | -0.009 | -0.008 | 2.8171 | 0.728 |
| | | 6 | -0.173 | -0.300 | 3.9618 | 0.682 |
| | | 7 | 0.188 | 0.212 | 5.3686 | 0.615 |
| | | 8 | 0.324 | 0.196 | 9.7787 | 0.281 |
| | | 9 | -0.273 | -0.162 | 13.082 | 0.159 |
| | | 10 | -0.067 | -0.024 | 13.293 | 0.208 |
| | | 11 | 0.180 | 0.151 | 14.887 | 0.188 |
| | | 12 | -0.060 | -0.001 | 15.077 | 0.237 |

* the results of Ljung-Box Q test of variable DTR

(b) DTR

Sample (adjusted): 1993 2020
Q-statistic probabilities adjusted for 2 dynamic regressors

| Autocorrelation | Partial Correlation | | AC | PAC | Q-Stat | Prob |
|---|---|---|---|---|---|---|
| | | 1 | -0.078 | -0.078 | 0.1905 | 0.662 |
| | | 2 | -0.127 | -0.134 | 0.7083 | 0.702 |
| | | 3 | 0.044 | 0.023 | 0.7739 | 0.856 |
| | | 4 | 0.015 | 0.004 | 0.7822 | 0.941 |
| | | 5 | 0.001 | 0.012 | 0.7822 | 0.978 |
| | | 6 | 0.081 | 0.086 | 1.0357 | 0.984 |
| | | 7 | 0.175 | 0.196 | 2.2599 | 0.944 |
| | | 8 | -0.110 | -0.058 | 2.7641 | 0.948 |
| | | 9 | -0.220 | -0.210 | 4.9093 | 0.842 |
| | | 10 | 0.046 | -0.035 | 5.0086 | 0.891 |
| | | 11 | 0.034 | -0.015 | 5.0665 | 0.928 |
| | | 12 | -0.097 | -0.099 | 5.5587 | 0.937 |

* the results of Ljung-Box Q test of variable ITR

(c) ITR

**Fig 1. Performance of residual autocorrelation diagnosis, 1991–2020.** *The Ljung Box verification results can be seen from Fig 1(a)-1(c), the Ljung Box test statistics from Phase 1 to Phase 12 were all 5% higher than the significant level, indicating that there was no self correlation between the residual items of Phase 1 and Phase 12.

**Table 4. Implementation of the residual heterogeneity test, 1991–2020, Taiwan.**

| Breusch Pagan Godfrey test | Model 1 | Model 2 | Model 3 |
|---|---|---|---|
| F-statistic | 0.162832 (0.6896) | 0.09177 (0.7642) | 0.212240 (0.6486) |
| OBS* R-squared | 0.173454 (0.6771) | 0.098004 (0.7542) | 0.225689 (0.6347) |
| Scaled explained SS | 0.464775 (04954) | 0.076161 (0.7826) | 0.991503 (0.3194) |

*The regression equations of model 1, model 2 and model 3 are derived from Table 4

means the first-order difference passes the stability test at 1% significance level, ** means the first-order difference passes the stability test at 5% significance level.

Further, in model 1 of Table 5, we discuss solely the nonlinear relation between TTR-to-GDP, where variable $GDP^2$ represents GDP squared, variable $GDP^3$ denotes GDP tripled. Including variables GDP, $GDP^2$, $GDP^3$, debt and consumer price index (CPI), all data are denominated in million TWD. Further, we establish the correlation among TTR, GDP, square GDP, triple GDP, debt and CPI as follows:

$$\Delta Totaltaxrevenue_t$$
$$= a_1 + \beta_{1i}GDP_t + \beta_{2i}(GDP_t)^2 + \beta_{3i}(GDP_t)^3 + \beta_{4i}Debt_t + \beta_{5i}CPI_t + \varepsilon_t \quad (10)$$

where $\varepsilon_t = \varphi_1\varepsilon_{t-1} + \varphi_2\varepsilon_{t-2} + \sigma_t$

Case 1: Eq (10) declares that TTR increases with the increase of GDP, reaching a significance of 10%, see model 1 of Table 5. That is, as the debt variable is included, TTR also increases with the increase of GDP, reaching a significance of 10%. However, as the consumer price index(CPI) variable is added. TTR also increases with the increment of GDP, but it does not reach the significance of 10%. In Table 6, we denote that TTR-to-GDP represents N-shaped curve relationship.

$$\Delta Directtaxrevenue_t$$
$$= a_1 + \beta_{1j}GDP_t + \beta_{2j}(GDP_t)^2 + \beta_{3j}(GDP_t)^3 + \beta_{4j}Debt_t + \beta_{5j}CPI_t + \varepsilon_t \quad (11)$$

where $\varepsilon_t = \varphi_1\varepsilon_{t-1} + \varphi_2\varepsilon_{t-2} + \sigma_t$

Case 2: Eq (11) demonstrates that DTR increases with the increment of GDP, see model 2 of Table 5, denoting the corresponding regression coefficient is 0.061, depicting the increment

**Table 5. Estimation results of TTR/DTR/ITR-to-GDP.**

| Dependent Variable: Total Tax revenue | GDP | (GDP)² | (GDP)³ | Debt | CPI | AR(1) | AR(2) | TSLS-(J-statistic) | DW |
|---|---|---|---|---|---|---|---|---|---|
| Model 1 | 0.3702 (1.6259) | -3.08E-8 (-1.7052) | 9.59E-16* (1.9862) | 0.0490 (0.2212) | 1127552 (0.6206) | 0.0033 | 0.0943 | 0.0000 | 2.0858 |
| Model 2 | 0.0610 (0.5958) | -1.52E-09 (-0.1726) | 9.13E-17 (0.3869) | -2.68E-16 (-0.1386) | 8.91E-11 (0.1601) | 0.0007 | 0.4486 | 0.0000 | 1.7011 |
| Model 3 | 0.3148** (2.5392) | -2.56E-8** (-2.6105) | 7.22E-16** (2.7444) | 0.1104 (0.9142) | 32265 (0.0326) | 0.0424 | 0.0107 | 7.66E-43 | 2.2459 |

[1]. In brackets is the t-statistic of the estimated parameter.

[2]. the GDP in 2020 is 19766240 measured in 100 million TWD.

[3]. The table is based on the historical data of the National Bureau of statistics of Taiwan.

[4]. Robust standard errors in parentheses.

$p^* < 0.10$, $p^{**} < 0.05$, $p^{***} < 0.01$.

**Table 6. Estimation of Tax Kuznets curve of Tax-to-GDP.**

| Year 2020 | Official GDP | Actual TR | TKC approach Tax evasion | Tax evasion/GDP | Tax-to-GDP Curve shaped |
|-----------|-------------|-----------|--------------------------|-----------------|-------------------------|
| TTR | 19766240 | 2398667 | 190522 | 0.009638 | N-shaped |
| DTR | 19766240 | 1324208 | 134050 | 0.006781 | N-shaped |
| ITR | 19766240 | 1074459 | 43248 | 0.002187 | N-shaped |

Note: The data sources of GDP, TTR, DTR, ITR in Table 5 are derived from the data of DGBAS, Taiwan

of GDP, to a certain extent, resulting in the increase of DTR. However, the coefficient does not pass the 10% significance test. Further, if the variables debt and CPI are added to the model. It shows that DTR increases with the growth of GDP, whereas these two coefficients fail within the significance test of 10%. In Table 6, we show that DTR-to- GDP presents N-shaped curve relationship.

$$\Delta Indirect\,tax\,revenue_t$$
$$= a_1 + \beta_{1k}GDP_t + \beta_{2k}(GDP_t)^2 + \beta_{3k}(GDP_t)^3 + \beta_{4k}Debt_t + \beta_{5k}CPI_t + \varepsilon_t \ (12)$$

where $\varepsilon_t = \varphi_1\varepsilon_{t-1} + \varphi_2\varepsilon_{t-2} + \sigma_t$

Case 3: Eq (12) illustrates that ITR increases with the increase of GDP, see model 3 of Table 5, reaching a significance of 1%. Even though variable debt is included, ITR also increases with the increment of GDP. Moreover, as variable CPI is added. ITR also increases with the increment of GDP. From the results of Table 6, our empirical research depicts that on the basis of the existing ITR-to- GDP, adding variable debt or CPI, the relationship between ITR-to-GDP presents a N-shaped relationship.

Obviously, in Table 5, we show that under low inflation, the promotion effect of consumer price index(CPI) upon total tax is not significant, our empirical result is in line with Khan et al. [51] argument.

## 4.2. Kuznet tax curve analysis (TTR/DTR/ITR and GDP)

Further, according to the statistics of DGBAS, Taiwan. Taiwan's GDP in 2020 is 19,766,240 million TWD, and the actual total tax revenue is 2,398,667 million TWD. However, according to Tax Kuznets curve of TTR-to-GDP ratio, when Taiwan's GDP in 2020 is 19,766,240 million TWD, the TTR should be 2,589,189 million TWD, revealing the total tax base evasion amount is 190,522 million TWD, accounting for 0.009638 of GDP in 2020. Our empirical results declare that Taiwan's total tax evasion rate in 2020 is 0.9638% (see Fig 2).

Similarly, Taiwan's GDP in 2020 is 19,766,240 million TWD, and the actual direct tax revenue is 1,324,208 million TWD. However, according to Tax Kuznets curve approach of DTR-to-GDP ratio, when Taiwan's GDP in 2020 is 1,324,208 million TWD, the DTR should be 1,458,258 million TWD, revealing that direct tax evasion is 134,050 million TWD, accounting for 0.6781 percent of GDP in 2020. Our empirical results depict that Taiwan's direct tax base evasion rate in 2020 is 0.6781% (see Fig 3).

Furthermore, Taiwan's GDP in 2020 is 19,766,240 million TWD, and the actual indirect tax revenue is 1,074,459 million TWD. However, according to Tax Kuznets curve of ITR to GDP ratio, when Taiwan's GDP in 2020 is 19,766,240 million TWD, the ITR should be 1,117,707 million TWD, revealing the indirect tax evasion is 43,248 million TWD, accounting for 0.002187 of GDP in 2020. Clearly, our empirical results demonstrate that Taiwan's indirect tax base evasion rate in 2020 is 0.2187% (see Fig 4). It can be seen from the above analysis that the tax base erosion rate of indirect tax is lower than that of direct tax. The main reason may be

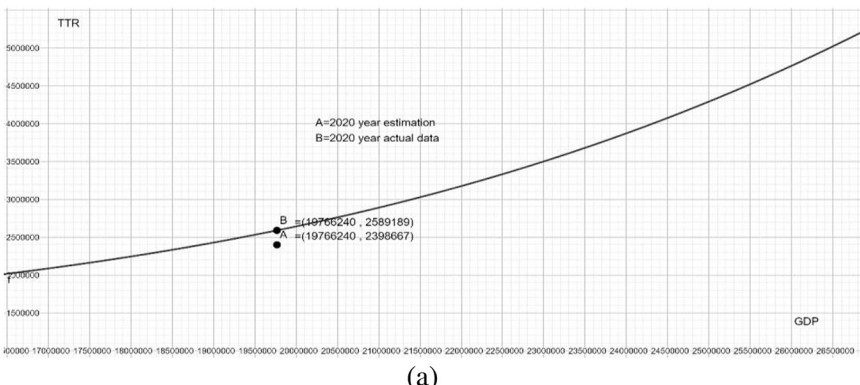

(a)

Point A is 2020 year estimation, point B is 2020 year actual data, where A means GDP is 19,766,240 million TWD and TTR is 2,398,667 million TWD. Point B means GDP is 19,766,240 million TWD and TTR is 2,589,189 million TWD.

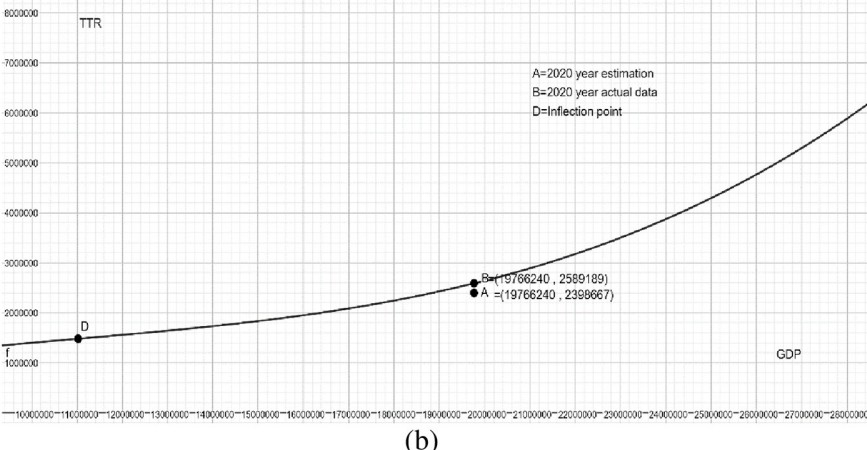

(b)

Point D means 2020 year inflection point

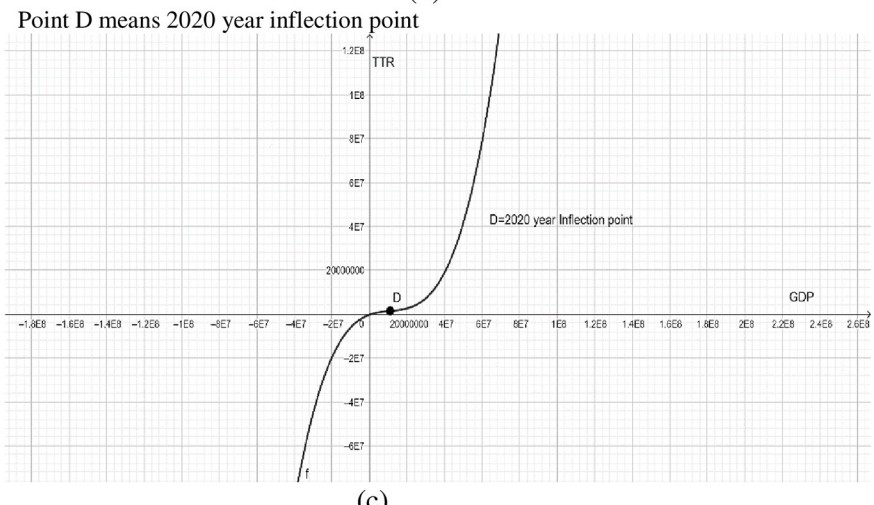

(c)

Separate display of TTR inflection point D in 2020

**Fig 2. TTR-to-GDP, Taiwan, 1991–2020.**

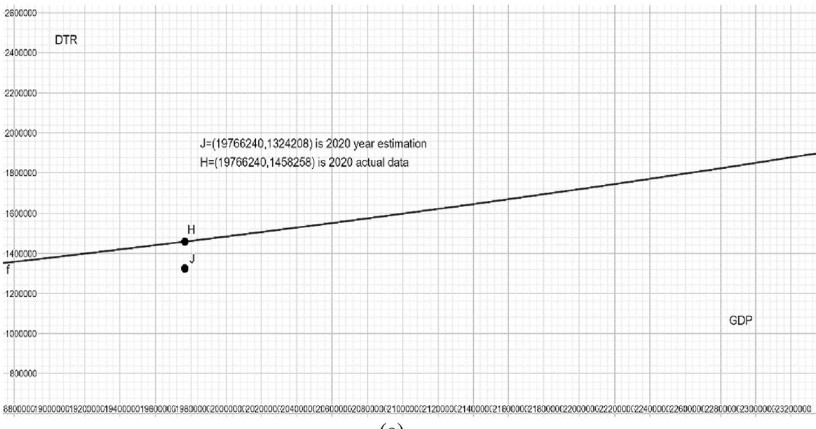

(a)

Point J is 2020 year estimation, point H is 2020 year actual data, where J means GDP is 19,766,240 million TWD and DTR is 1,324,208 million TWD. Point H means GDP is 19,766,240 million TWD and DTR is 1,458,258 million TWD.

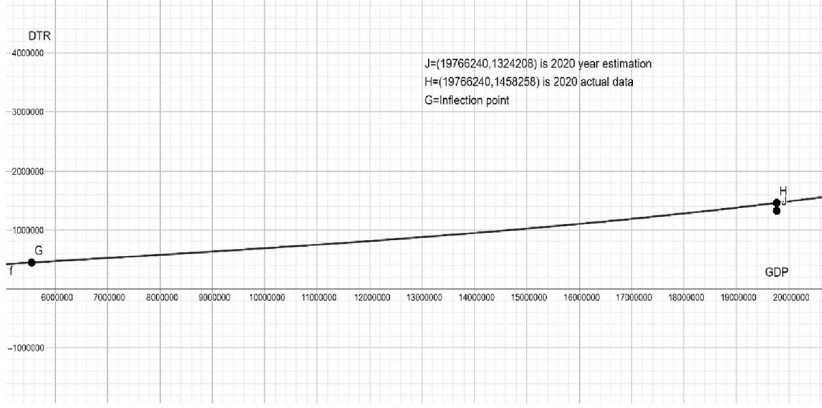

(b)

Point G means 2020 year inflection point

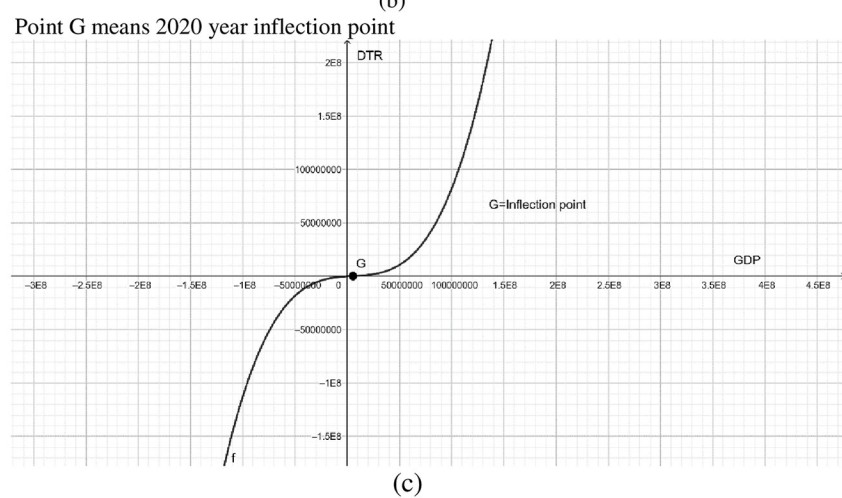

(c)

Separate display of DTR inflection point G in 2020

**Fig 3. DTR-to-GDP, Taiwan, 1991–2020.**

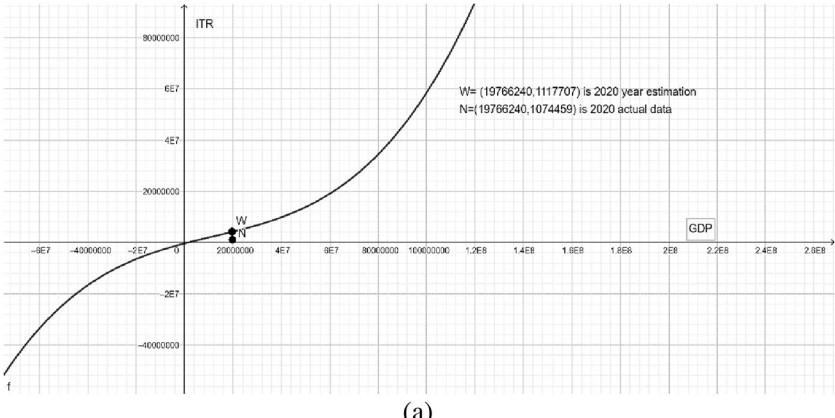

(a)

Point W is 2020 year estimation, point N is 2020 year actual data, where W means GDP is 19,766,240 million TWD and ITR is 1,117,707 million TWD. Point N means GDP is 19,766,240 million TWD and ITR is 1,074,459 million TWD.

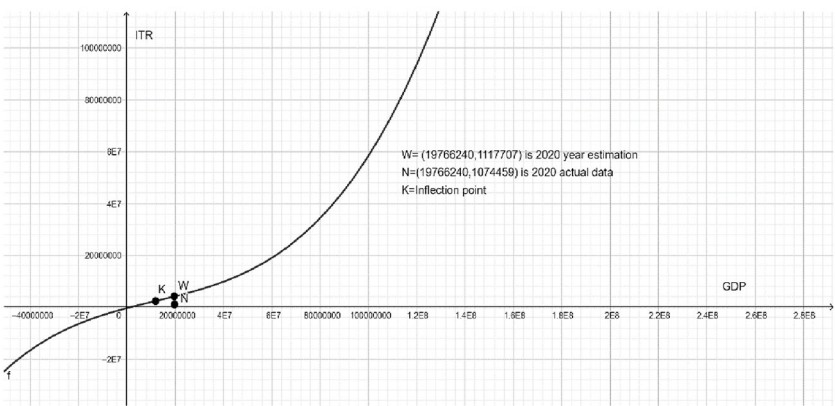

(b)

Point K means 2020 year inflection point

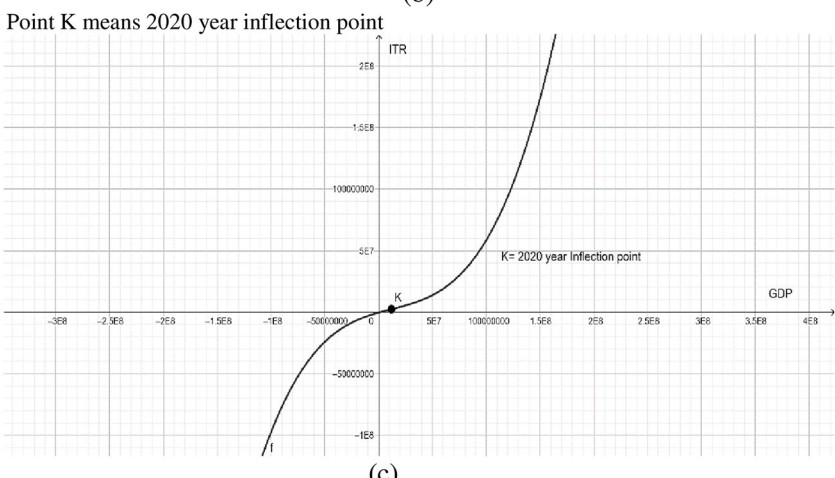

(c)

Separate display of ITR inflection point K in 2020

**Fig 4. ITR-to-GDP, Taiwan, 1991–2020.**

that indirect tax is levied by withholding at source, which is difficult to evade taxpayment for taxpayers.

## 4.3. Discussion and recommendation

In this paper, we take Taiwan as a case study and use Simultaneous equations model and SUR-OLS approach to exploit the relationship among the GDP, TTR, DTR, ITR for Taiwan over a time period ranging from 1991 to 2020. The difference with Wang et al. [12] is that this paper uses the data from the Taiwan General Accounting Office database to bring into the theoretical model established in this paper and uses empirical analysis methods to estimate the amount of tax base erosion of total tax/direct tax/indirect tax in 2020, and compares the rate of tax base erosion to find that the rate of tax base erosion of indirect tax is smaller than the rate of tax base erosion of direct tax, this section adopts the Tax Kuznet Curve analysis, and the results obtained are consistent with those obtained in the next section according to the threshold theory analysis (indirect taxes have no threshold effect, but direct taxes have threshold effect). Our research opens up a new path for the research of TTR/DTR/ITR to GDP, and also fills in the theoretical gaps on these issues The findings and implications will offer indicative guideline for the study of the relationship among TTR/DTR/ITR and GDP.

## 5. Threshold analyses

### 5.1. Empirical analyses

To explore whether Taiwan's tax burden rate has a threshold value that changes the relationship between the tax burden rate and the economic growth of the underground sector. Different from Lin et al. [42], this paper, after controlling the influence of other variables (excluding tax burden rate), discusses how GDP and (total tax/direct tax/ indirect tax) fluctuate between 1991 and 2020, and measures whether there exists a threshold effect among TTR/DTR/ITR and GDP. We follow the threshold regression model of Bai and Perron [43], and take tax burden rate as the threshold variable, Check whether the tax burden rate has different effects on Taiwan's total tax revenue/direct tax revenue/indirect tax revenue in different high and low intervals under the control of other control variables, including government debt and consumer price index. Owing to different individuals have different thresholds, it is necessary to emphasize the determinants of threshold. Eq (13) is the list and definition of variables that being used to test the taxation threshold effect on economic growth. The model is written as follows:

$$
(TTR/DTR/ITR)_{i,t} = \mu_i + \theta_1' GDP_{i,t} I(q_{i,t} \leq \check{t}_1) + \theta_2' GDP_{i,t} I(\check{t}_1 < q_{i,t} \leq \check{t}_2) + \theta_3' GDP_{i,t} I(q_{i,t} > \check{t}_2) \\
+ \alpha Debt_{i,t} + \beta CPI_{i,t} + \varepsilon_{i,t}
$$

(13)

where I (E) is an indicator function. When event E occurs, I (E) = 1, otherwise I (E) = 0, the residual term $e_t = [e_{1,t}, e_{2,t}]$, $\theta_1'$, $\theta_2'$, $\alpha$, $\beta$ are parameters to be estimated, $\check{t}$ is threshold tax variable, $(TTR/DTR/ITR)_{i,t}$ is the explained variable, $Debt_{i,t}$ and $CPI_{i,t}$ are explanatory variables, and $T \equiv [t_0, t_1]$ is the spatial parameter of $\check{t}_1$ $\check{t}_2$, $\check{t} \in T$. The threshold variable $q_{i,t}$ is either smaller or larger than the threshold $\check{t}_1$ that illustrate by slopes $\theta_1'$, $\theta_2'$ and $\theta_3'$. I(·) is the indicator function, which takes the value 1 if the argument in parenthesis is valid, and 0 otherwise. The $\varepsilon_{i,t}$ is assumed to be identically and independently distributed (iid) with mean equal to zero and variance is finite, that is $e_{it} \approx [0 \sim \sigma^2]$.

In this study, we set the dependent variable as *TTR/DTR/ITR* and use the income square term and income cubic term as explanatory variables to capture the nonlinear impact relationship between these variables and total tax/direct tax/indirect tax Hansen [34]. Owing to over

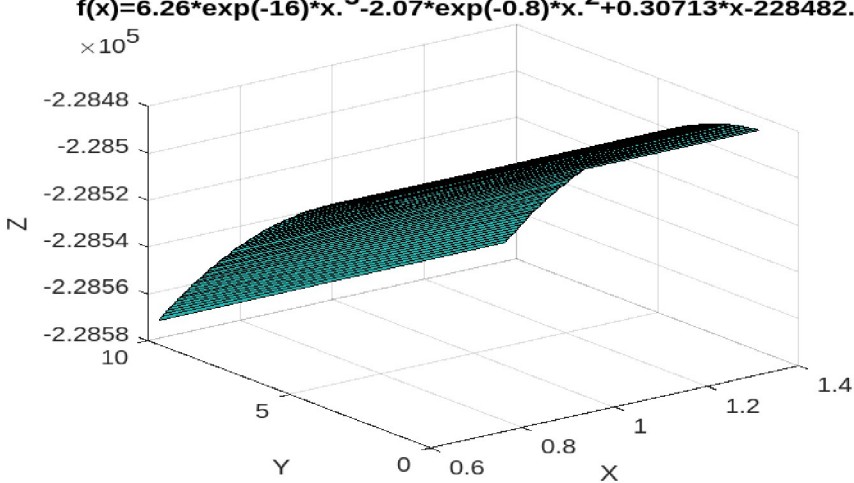

**Fig 5. TTR-to-GDP threshold analyses, Taiwan, 1991–2020.**

parameterization will also reduce the degree of statistical freedom, resulting in inefficient regression estimation results. Hence, our research selects the tax rate as the threshold parameter. The empirical results can be obtained as follows:

Case 1: TTR-to-GDP

According to Eqs (10) and (13). When the tax burden rate is below 12.5%, the increase in GDP at this stage will produce a positive effect on total tax revenue. When the tax burden rate is between 12.5% and 13%, at this stage, the increase in GDP causes the total tax revenue to fall instead of increasing, indicating that the total tax base is being eroded. However, when the tax burden rate is greater than 13%, the increase in GDP at this stage will have a positive effect on total tax revenue. That is, the relationship between GDP and TTR presents a N-shaped relationship (see Fig 5).

Case 2: DTR-to-GDP

Similarly, according to Eqs (11) and (13). When the tax burden rate is below 12.6%, the increase in GDP at this stage will produce a positive effect on direct tax revenue. However, when the tax burden rate is between 12.6% and 13.4%, at this stage, the increase in GDP causes the direct tax revenue to fall instead of increasing, indicating that the direct tax base is eroding. Moreover, when the tax burden rate is greater than 13.4%, the increase in GDP at this stage will have a positive effect on direct tax revenue. That is, the relationship between GDP and direct tax revenue presents a N-shaped relationship (see Fig 6).

Case 3: ITR-to-GDP

Next, according to Eqs (12) and (13), we take the tax burden rate as the threshold variable, our empirical result reveals that indirect tax has no threshold effect, that is, with the increase of GDP, indirect tax revenue also increases. Clearly, the relationship between GDP and indirect tax revenue demonstrates an $\Gamma$-shaped relationship (see Fig 7).

## 5.2. Discussion and recommendation

Our empirical research can be summarized as follows:

(i) When the total tax rate is below 12.5%, taxpayers are willing to pay even if the tax rate increases because the "expected benefit" of tax evasion is less than the penalty cost of being

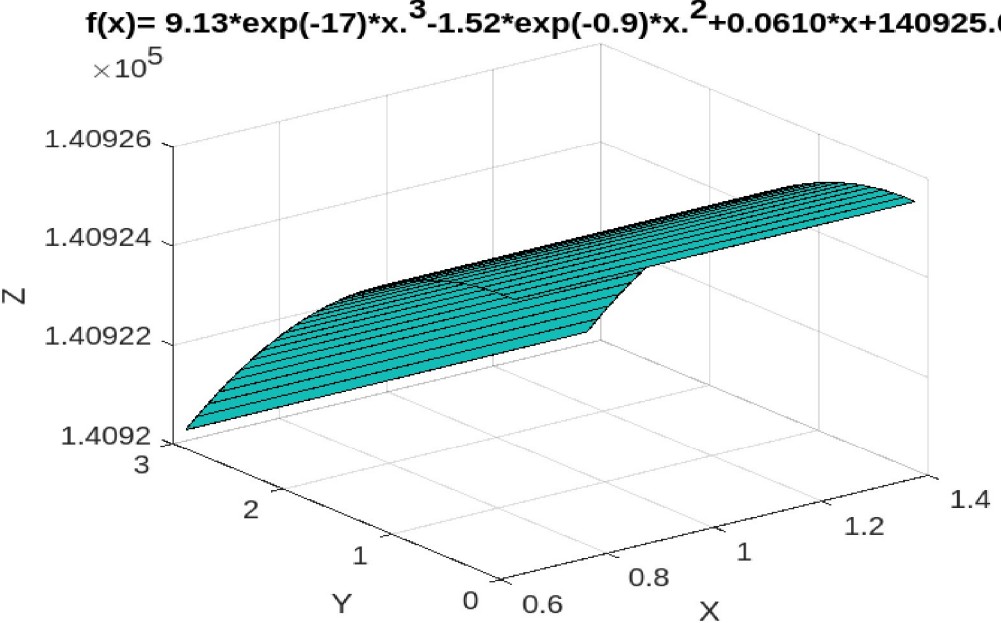

$$f(x)= 9.13*exp(-17)*x.^3-1.52*exp(-0.9)*x.^2+0.0610*x+140925.6$$

**Fig 6. DTR-to-GDP threshold analyses, Taiwan, 1991–2020.**

caught. However, when the total tax rate is between 12.5% and 13%, taxpayers measure the "expected benefits" of tax evasion to outweigh the penalty costs of being caught. So total tax revenue in this range will decrease as GDP grows, not increase. In fact, according to statistics from Taiwan's Ministry of Finance, the actual average total tax rate from 1991 to 2020 was 12.7%, which is between the threshold total tax rate of 12.5% and 13%, indicating that within this tax rate range, there is tax base erosion in total tax revenue. As mentioned earlier in this

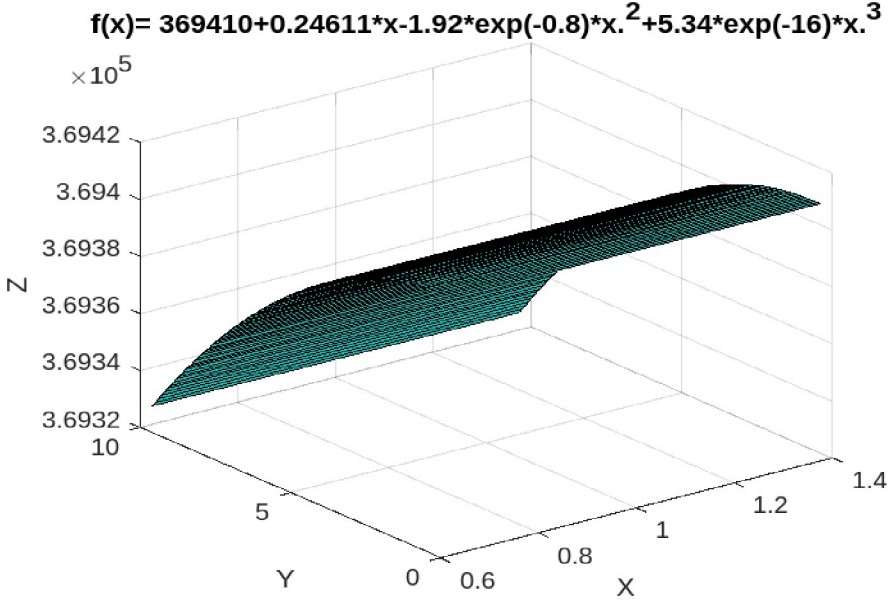

$$f(x)= 369410+0.24611*x-1.92*exp(-0.8)*x.^2+5.34*exp(-16)*x.^3$$

**Fig 7. ITR-to-GDP threshold analyses, Taiwan, 1991–2020.**

article, according to statistics from Taiwan's DGBAS. In 2020, Taiwan's GDP is NT$19,766.24 billion, and its actual tax revenue is NT$239.8667 billion. However, according to the tax Kuznets ratio curve (TTR-to-GDP), when Taiwan's GDP in 2020 is 197.6624 billion TWD, TTR should be 258.9189 billion TWD, showing that the total tax evasion in 2020 is 190.522 billion TWD, accounting for 0.009638 of GDP, which is consistent with the empirical results of the threshold theory mentioned above. (ii) When the direct tax rate is lower than 12.6%, taxpayers are still willing to pay taxes even if the tax rate is raised because the "expected benefit" of taxpayer evasion is less than the penalty cost of being caught. However, when the average direct tax rate is between 12.6% and 13.4%, taxpayers measure the "expected benefits" of tax evasion outweighing the penalty costs of being caught. Consequently, direct tax revenues within this threshold will decrease rather than increase as the economy grows. Our empirical results show that the tax base erosion rate of Taiwan's total tax revenue in 2020 is 0.9638%, which is consistent with our estimated results based on the threshold model. (iii) Finally, our empirical results show that from 1991 to 2020, Taiwan's indirect tax does not have a threshold effect, that is, indirect tax revenue increases with the growth of GDP.

## 6. Conclusion

Our research differs from the traditional methodology, we adopt SUR-OLS method, Tax Kuznet Curve (TKC) approach and Threshold model to estimate the response of GDP on total tax revenue(TTR), direct tax revenue(DTR) and indirect tax revenue (ITR) in Taiwan from 1991–2020. In empirical research, we select the parameter "tax burden rate" as a threshold variable to capture the response of income variation upon tax revenue. Our research contributed to the literature on Threshold analysis regarding the optimal tax rate and tax erosion as follows: First, according to the Kuznet Tax curve (KTC) approach, we find that Taiwan's total tax base erosion rate in 2020 is 0.9638%, indicating that the tax base erosion rate in Taiwan is not so severe. Our research show the empirical results of Threshold model and Kuznet Tax Curve approach are consistent. Second, our empirical result shows that total tax and direct tax have threshold effect, but indirect tax has no threshold effect, which reflects that the difference of tax structure will affect the robustness of empirical results when conducting empirical research on tax burden rate and tax base erosion. Third, according to the Kuznet Tax Curve model, we estimate the indirect tax base erosion rate in 2020 is 0.2187%, which is lower than the direct tax base erosion rate of 0.6187%. Obviously, our empirical research shows that indirect tax revenue without "tax rate threshold effect" is more effective in reducing the tax base erosion rate than direct tax with tax rate threshold effect. This reflects that the difference of tax structure between direct tax and indirect tax plays a key role in the empirical study of tax burden rate and tax base erosion. Finally, threshold models are widely used in economics. However, there are limitations to assuming that the threshold is stable or time-invariant. this paper adopts a piece wise in variable analysis method, not a piece wise in time analysis method, if "time" is taken as the threshold variable, its significance is to analyze the time point before and after the structural change of the tax burden rate. In the future, we can consider further examining the impact of fiscal policy change on taxation by taking "time" as a threshold variable.

## Supporting information

**S1 File.**
(DOCX)

**S2 File.**
(DOCX)

**S3 File.**
(DOCX)

**S4 File.**
(DOCX)

**S1 Data.**
(XLSX)

## Acknowledgments

We are especially grateful to the referees of PLOS ONE for their comments, as well as to all members of the Zhanjiang University of Science and Technology and Chinese Culture University for their continuous aid and support during the completion of this research.

## Author Contributions

**Conceptualization:** Yu kun Wang.

**Data curation:** Yu kun Wang.

**Formal analysis:** Yu kun Wang, Li Zhang.

**Funding acquisition:** Yu kun Wang.

**Investigation:** Yu kun Wang.

**Methodology:** Yu kun Wang.

**Project administration:** Yu kun Wang.

**Resources:** Yu kun Wang, Li Zhang.

**Software:** Yu kun Wang.

**Supervision:** Yu kun Wang, Li Zhang.

**Validation:** Yu kun Wang.

**Visualization:** Yu kun Wang.

**Writing – original draft:** Yu kun Wang.

**Writing – review & editing:** Yu kun Wang.

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
