## [Decision Letter · Decision Letter 0]

23 Nov 2022

PONE-D-22-29289Optimal tax rate and tax evasion; threshold analysis from Taiwan Yu kun Wang1 Li Zhang2PLOS ONE

Dear Dr. Wang,

Thank you for submitting your manuscript to PLOS ONE. After careful consideration, we feel that it has merit but does not fully meet PLOS ONE’s publication criteria as it currently stands. Therefore, we invite you to submit a revised version of the manuscript that addresses the points raised during the review process.

We look forward to receiving your revised manuscript.

Kind regards,

Rana Muhammad Ammar Zahid, PhD

Academic Editor

PLOS ONE

Journal Requirements:

"The author(s) received no specific funding for this work"

5. Please amend your list of authors on the manuscript to ensure that each author is linked to an affiliation. Authors’ affiliations should reflect the institution where the work was done (if authors moved subsequently, you can also list the new affiliation stating “current affiliation:….” as necessary).

6. We note you have included a table to which you do not refer in the text of your manuscript. Please ensure that you refer to Table 1, 2 and 4 in your text; if accepted, production will need this reference to link the reader to the Table.

**Additional Editor Comments:**

The theme of the paper is interesting. Please incorporate the changes suggested by the reviewers.

Reviewers' comments:

Reviewer's Responses to Questions

**Comments to the Author**

1. Is the manuscript technically sound, and do the data support the conclusions?

Reviewer #1: Yes

Reviewer #2: Yes

2. Has the statistical analysis been performed appropriately and rigorously? 

Reviewer #1: Yes

Reviewer #2: Yes

3. Have the authors made all data underlying the findings in their manuscript fully available?

Reviewer #1: Yes

Reviewer #2: Yes

4. Is the manuscript presented in an intelligible fashion and written in standard English?

Reviewer #1: Yes

Reviewer #2: No

5. Review Comments to the Author

Reviewer #1: Review report

of the article:

Optimal tax rate and tax evasion; threshold from Taiwan

I have for the authors the following recommendations for improving the manuscript:

1. I recommend the title modification: Threshold analysis regarding the optimal tax rate and tax evasion. Empirical evidence from Taiwan

2. The introductory part must encompass the research's scope and objectives and the paper's main added value regarding the research questions.

3. The literature review section is missing. There are titles and citations in the introductory part, but the current state of the art must be inserted and expanded. It will be interesting if the authors can introduce a table with the main similar empirical research outcomes both in Taiwan and at the international level.

4. The model part is interesting but must also encompass the research hypothesis. Without any research questions, there is no connectivity between theory and practice.

5. The name and implications must be inserted in the Kuznets Tax Curve Analysis under each subgraph for better and easier identification by the audience.

6. I recommend a discussion and a recommendation section where the authors will debate and analyze the current research paths with the ones that have been debated in the empirical, studies literature.

7. The conclusions section must be expanded with the limitations of the study and with the path for the upcoming future research developments in the field for the authors.

8. The reference section must be seriously improved. The references are quite old and must be updated. The last quoted title is from 2018. There must be updated the literature review section with 10-15 titles from the 2019-2022 period.

Reviewer #2: Manuscript Number: PONE-D-22-29289

Title: Optimal tax rate and tax evasion; threshold analysis from Taiwan

This study explains the relationship among economic growth, tax rate, debt, consumer price index and tax revenue. This study differs from the traditional methodology. Authors adopt SUR-OLS method and Threshold approach to estimate the response of economic growth on total tax revenue, direct tax revenue and indirect tax revenue in Taiwan from 1991- 2020. This paper further discusses the impact of total, direct, and indirect taxes on tax rate volatility.

My detail comments are as following.

The format of manuscripts is not according to PLOS ONE requirement. Format of Manuscript need a comprehensive revision. Quality of tables and figures is poor. Carefully add table and figure numbers. There are too many subsections better follow PLOS ONE template to revise the paper.

Abstract: The section is poorly written. The abstract section should clearly mention the objective, methods, and concise results with sequence. Revise accordingly and add all parts, exclude unnecessary sentences and words in this section. Be specific

Introduction: In the introduction section, research objectives are missing. Clarify the background, objectives, research gaps, and innovations in this section. There are too many citations in this section better add an additional section of literature review and include more advance and relevant literature related to tax rate and tax evasion.

Although methods are advanced and appropriate; however, add the relevance and importance of these methods to justify the application in this study. Further add methods and data section in the manuscript

Better add Results and Discussion section: Discuss results in detail and mention the innovative outcome. Further results should be backed with appropriate literature. Results should be further elaborated in detail.

Conclusion: Add limitations and future research ideas in the conclusion section. Expand the policy implications for Taiwan.

References: References are not aligned with Journal Format, Revise it

Avoid grammatical and typo errors and revise the manuscripts for these two concerns.

6. PLOS authors have the option to publish the peer review history of their article (what does this mean?). If published, this will include your full peer review and any attached files.

Reviewer #1: No

Reviewer #2: No

---

## [Author Response · Author response to Decision Letter 0]

5 Jan 2023

We are especially grateful to the academic editor and reviewer(s) of PLOS ONE for their comments, as well as to all members of the Editorial Board for their giving us continuous aid and support during the completion of this research.

---

## [Decision Letter · Decision Letter 1]

16 Jan 2023

Threshold analysis regarding the optimal tax rate and tax evasion. empirical evidence from Taiwan

PONE-D-22-29289R1

Dear Dr. Wang,

We’re pleased to inform you that your manuscript has been judged scientifically suitable for publication and will be formally accepted for publication once it meets all outstanding technical requirements.

Kind regards,

Rana Muhammad Ammar Zahid, PhD

Academic Editor

PLOS ONE

Additional Editor Comments (optional):

Reviewers' comments:

Reviewer's Responses to Questions

**Comments to the Author**

1. If the authors have adequately addressed your comments raised in a previous round of review and you feel that this manuscript is now acceptable for publication, you may indicate that here to bypass the “Comments to the Author” section, enter your conflict of interest statement in the “Confidential to Editor” section, and submit your "Accept" recommendation.

Reviewer #1: All comments have been addressed

Reviewer #2: All comments have been addressed

2. Is the manuscript technically sound, and do the data support the conclusions?

Reviewer #1: Yes

Reviewer #2: Yes

3. Has the statistical analysis been performed appropriately and rigorously? 

Reviewer #1: Yes

Reviewer #2: Yes

4. Have the authors made all data underlying the findings in their manuscript fully available?

Reviewer #1: Yes

Reviewer #2: Yes

5. Is the manuscript presented in an intelligible fashion and written in standard English?

Reviewer #1: Yes

Reviewer #2: Yes

6. Review Comments to the Author

Reviewer #1: The authors have addressed all issues that I have raised. Now, the manuscript is improved and looks much better. the scientifical soundness of the research has been improved as also the Methodology, results, conclusions, and References.

Reviewer #2: (No Response)

7. PLOS authors have the option to publish the peer review history of their article (what does this mean?). If published, this will include your full peer review and any attached files.

Reviewer #1: No

Reviewer #2: No

---

## [Editor Report · Acceptance letter]

19 Jan 2023

PONE-D-22-29289R1 

Threshold analysis regarding the optimal tax rate and tax evasion. empirical evidence from Taiwan 

Dear Dr. Wang:

I'm pleased to inform you that your manuscript has been deemed suitable for publication in PLOS ONE. Congratulations! Your manuscript is now with our production department. 

Kind regards, 

on behalf of

Dr. Rana Muhammad Ammar Zahid 

Academic Editor

PLOS ONE